# Spray Coating of Wood with Nanoparticles from Lignin and Polylactic Glycolic Acid Loaded with Thyme Essential Oils

**DOI:** 10.3390/polym16070947

**Published:** 2024-03-30

**Authors:** Florian Zikeli, Jasmina Jusic, Cleofe Palocci, Giuseppe Scarascia Mugnozza, Manuela Romagnoli

**Affiliations:** 1Department for Innovation in Biological, Agro-Food and Forest Systems (DIBAF), University of Tuscia, 01100 Viterbo, Italy; zikeli@unitus.it (F.Z.); jasmina.jusic@unitus.it (J.J.); gscaras@unitus.it (G.S.M.); 2Fraunhofer, Via Alessandro Volta 13A, 39100 Bozen, Italy; 3Department of Chemistry, Sapienza University of Rome, P.le A. Moro 5, 00185 Rome, Italy; cleofe.palocci@uniroma1.it; 4Research Center for Applied Sciences to the Safeguard of Environment and Cultural Heritage (CIABC), Sapienza University of Rome, P.le A. Moro 5, 00185 Rome, Italy

**Keywords:** lignin nanoparticles, essential oils, spray coating, PLGA, FTIR imaging, SEM

## Abstract

Ensuring the longevity of wooden constructions depends heavily on the preservation process. However, several traditional preservation methods involving fossil-based compounds have become outdated because they pose a significant risk to the environment and to human health. Therefore, the use of bio-based and bioactive solutions, such as essential oils, has emerged as a more sustainable alternative in protecting wood from biotic attacks. The entrapment of essential oils in polymeric carrier matrices provides protection against oxidation and subsequent degradation or rapid evaporation, which implies the loss of their biocidal effect. In this work, lignin as well as PLGA nanoparticles containing the essential oils from two different thyme species (*Thymus capitatus* and *T. vulgaris*) were applied on beech wood samples using spray coating. The prepared coatings were investigated using FTIR imaging, SEM, as well as LSM analysis. Release experiments were conducted to investigate the release behavior of the essential oils from their respective lignin and PLGA carrier materials. The study found that lignin nanoparticles were more effective at trapping and retaining essential oils than PLGA nanoparticles, despite having larger average particle diameters and a more uneven particle size distribution. An analysis of the lignin coatings showed that they formed a uniform layer that covered most of the surface pores. PLGA nanoparticles formed a film-like layer on the cell walls, and after leaching, larger areas of native wood were evident on the wood samples treated with PLGA NPs compared to the ones coated with lignin NPs. The loading capacity and efficiency varied with the type of essential oil, while the release behaviors were similar between the two essential oil types applied in this study.

## 1. Introduction

In recent years, there has been a significant increase in interest regarding the use of wood as a construction material for the future. This is due to its ability to sequester carbon dioxide into woody tissue, as well as its potential to reduce the carbon footprint of new buildings by up to 50% when replacing concrete or steel [1,2,3]. Wood-based buildings face obstacles in gaining general acceptance due to the risk of degradation by wood-rot fungi. This is especially true when using less durable wood species like beech, which is abundant in Europe and suitable for the bio-building sector [4,5]. Wood preservation is a necessity to prolong the service life of wood products and the current strategies have replaced toxic fossil-based chemical wood preservation agents with more environmentally friendly methods like heat treatments [6,7,8,9,10,11,12,13,14] or impregnation with natural compounds [15,16,17,18,19,20] or the utilization of naturally durable species [21,22,23,24]. 

Essential oils (EOs) have been extensively studied as natural bioactive compounds in recent years with interesting results when used to protect wood against the most widespread wood rots and bacteria [25,26,27,28,29,30,31,32,33,34,35,36,37,38,39]. Entrapment of EOs into drug delivery particles is a further step towards a more sustainable wood preservation approach, increasing the efficiency of EO through controlled release, which gradually achieves the desired outcome [39,40,41,42]. 

Other options to make wood protection more environmentally friendly is to replace fossil-based ingredients in traditional wood protection formulations with bio-based additives like lignin or to use lignin nanoparticles (LNPs) as coatings or as carriers of bioactives [37,43,44,45,46,47,48,49,50,51,52,53,54,55,56,57,58,59,60]. Lignin coatings were investigated in a wide range of applications [61,62,63], including as water vapor and oxygen barriers in fiber-based packaging solutions [64], as protective coatings for natural fibers for textile production [65], or for the surface modification of paper products [66,67]. Others prepared novel, bio-based anti-viral surface coatings via spin-coating of lignin where the anti-viral effect was obtained due to reactive oxygen species formation through lignin phenol oxidation and therefore, the lignin coating is not consumed [68].

In recent years, LNPs have found numerous applications as nano- or microstructured polymeric carrier matrices for biocide delivery systems in agriculture or pharma because of its UV-protective and antioxidant properties as well as its ability to protect against unwanted fast evaporation of volatile loadings [52,69,70,71,72,73,74,75,76,77,78,79]. However, the application of pure LNP wood conservation formulations are still very rare. Moreno et al. [80] utilized the extractive urushiol, a catechol derivative with an unsaturated hydrocarbon side-chain, as an internal cross-linker of co-aggregated LNPs. As a result, the LNPs were stabilized and hydrophobic LNP-based wood coatings were prepared upon cross-linking during thermal curing of urushiol–LNP aqueous dispersions on the wood surface. In earlier works, the authors of this study prepared LNPs with entrapped cinnamon and thyme EOs for the subsequent delayed release of the incorporated EOs, and the LNPs with entrapped *Thymus* spp. EO gave promising results for the protection of pine seedlings against *Phytophtora cactorum* as well as against wood-rot fungi [31,39,59].

Poly(d,l-lactic-co-glycolic) acid (PLGA) is a widely employed polymer that has been extensively studied as a polymeric delivery system for various bioactive molecules and has found wide use in pharmaceutical and agricultural applications [81,82,83,84]. Since there are bio-based production methods for both lactic acid and glycolic acid monomers, PLGA could be considered a fully biobased polymer [85,86,87]. PLGA is FDA approved, which makes it an attractive option for use in packaging and biomedical applications. Therefore, PLGA finds extensive utilization in the synthesis of biomaterials for bone tissue engineering [88]. One main advantage of PLGA micro- and nanoparticles is their effective drug retention, leading to a slow and continuous release of the incorporated drugs, which helps prolong the residence time of biocidal particles at the target site [84]. When using specialized devices like a microfluidic reactor, it was possible to prepare small PLGA nanoparticles (NPs) with particle diameters < 50 nm, which are necessary to penetrate plant cells in order to discharge the load at specific target sites like plastids, vacuoles, or cell nucleus [89]. By employing fluorescent markers incorporated into PLGA NPs, the uptake of PLGA NPs into Arabidopsis thaliana cells as well as plantlet roots was shown using fluorescent microscopy [90]. PLGA was also employed in coating applications like thin films with added plasticizers [91], and films blended with other biobased polymers such as PLA [92] or lignin [93], but literature reports about the use of PLGA NPs alone for the preparation of coatings were not found.

The aim of this study was to apply EO-containing LNPs (EOLs) as well as EO-containing PLGA NPs (EOPs) as surface treatments for beech wood samples using a spray-coating method, simulating a biocide application to a wood-based building. The subsequent release experiments of the wood samples immersed in water were designed to investigate the retention of the EOs and their leaching behavior from the coated wood samples in the case of high humidity. To the best of the authors’ knowledge, this is the first time that a study prepared wood coatings consisting of completely bio-based EOLs and EOPs, which were applied via spray coating.

## 2. Materials and Methods

### 2.1. Essential Oil-Containing Nanoparticle Preparation

Organsolv lignin (OSL) from beech wood was supplied by Fraunhofer CBP (Leuna, Germany). Acetone (HPLC grade) and LLG SPHEROS PES 0.22 µm syringe filter (Lab Logistics Group, Meckenheim, Germany) were purchased from Carlo Erba Reagents srl (Milano, Italy). MeOH (≥99.9%, HPLC grade) and poly(D,L)-lactic-co-glycolic acid (PLGA, lactide-glycolide 50:50, MW 30–60 kDa) were purchased from Sigma-Aldrich (Merck KGaA, Darmstadt, Germany). Spectra/Por^®^ dialysis membranes with an MWCO of 6–8 kDa were purchased from Carlo Erba Reagents srl (Milano, Italy).

Essential oils (EOs) from *Thymus capitatus* (TC) and *T. vulgaris* (TV), identical to the ones used in an earlier work [30], were kindly provided by Flora srl, Florence, Italy (Table 1). Flora srl extracted the respective essential oils by hydrodistillation using a Clevenger-type apparatus. The chemical compositions were determined by Flora srl, using a PerkinElmer Clarus 500 GC-FIDMS system (Waltham, MA, USA), and are reported in Table 1.

Essential oil-containing lignin nanoparticles (EOLs) were prepared according to the protocol reported in Zikeli et al. [39] with modifications. OSL (300 mg) and each EO (100 mg) were dissolved in 10 mL acetone and transferred into dialysis bags, which were exposed to an excess of distilled water (2 L) under stirring for 2.5 h at room temperature. The dialysis time was long enough to remove the acetone but short enough to prevent the release of the entrapped EO. After dialysis, the samples were kept in the fridge for further analysis and application. EO-containing PLGA nanoparticles (EOPs) were prepared using the same protocol as for the EOLs. The solid contents of EOLs and EOPs were determined by freeze-drying aliquots of the prepared NP dispersions. The EO content was determined by HPLC (Agilent 1100 series with diode array detector) using methanol and water as mobile phases for gradient elution and a Robusta C18 3u (150 mm length, 4.6 mm internal diameter) reversed-phase HPLC column (Sepachrom Srl, Rho, Italy). In particular, 400 µL of the respective NP dispersions were mixed with 600 µL MeOH, filtered through a 0.20 µm PES syringe filter, and injected into the HPLC system. Calibration was performed using dilutions of the pure EOs in methanol–water (60:40 vol%).

### 2.2. Spray Coating of Wood Samples

Beech wood samples with a size of 25 × 25 × 5 mm (length × width × height) were used for the spray-coating experiments. The dry weight of the wood samples was determined after drying at 105 °C in a ventilated GC oven until a constant weight was reached (HP 5890 Series II, Agilent Technologies, Santa Clara, CA, USA). The set-up for the experiments is illustrated in Figure 1. EOL (1) and EOP (2) dispersions were applied on wood samples (3), which were placed on a sample desk (5) inside a Plexiglas chamber (6) and covered by a Plexiglas plate (7), utilizing a simple portable nebulizer (4) (Nano Mist Sprayer, Shi Yan Xin Ke Li Ltd., Shenzhen, China). Similarly, aqueous dispersions of the pure EOs (EO-TC alone, EO-TV alone) were utilized for spray coating. The dispersions were prepared using a concentration of 5.1 mg/mL in the case of EO-TC alone and 4.3 mg/mL in the case of EO-TV alone. After vortexing for 15 min, the EO dispersions were used for the spray-coating experiments. The nebulizer was attached by Velcro tape to the chamber wall in such a way that the spray steam covered all wood samples placed on the sample desk. The volume of the respective NP dispersion used for one coating layer was 15 mL, and the spraying intervals of 1 min were interrupted by a 1 min pause. The wood samples before and after spray coating were dried in a ventilated GC oven at 40 °C for 2 h in order to not compromise the applied coating by evaporation of the EOs; the sample were weighed 15 min after they were taken out of the oven. Drying of the beech wood samples at 40 °C for 2 h resulted in a humidity of around 5% relative to the dry weight of the samples. The same methodology was used for the spray coating of the backside of the wood samples. Accordingly, the coating uptake was calculated using Equation (1):CU [%] = (w_2_ − w_1_)/w_1_(1)

CU—coating uptake in percent;w_1_—uncoated sample weight after drying at 40 °C for 2 h;w_2_—coated sample weight after drying at 40 °C for 2 h. 

**Scheme 1 polymers-16-00947-sch001:**
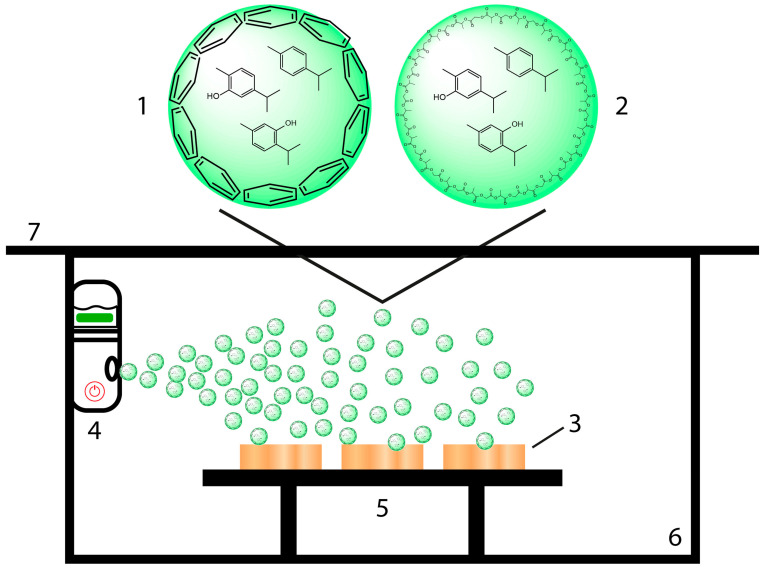
Illustrative set-up for spray coating using aqueous dispersions of lignin nanoparticles (EOLs, (1)) and PLGA nanoparticles loaded with essential oils (EOPs, (2)) on beech wood samples (3), using a portable nebulizer (4) and a simple sample desk (5) in a Plexiglas chamber (6) covered by a Plexiglas plate (7).

### 2.3. Release Experiments of EOs from Spray-Coated Wood Samples

The coated wood samples were placed in Petri dishes (5 cm diameter, one sample per Petri dish) and immersed in 10 mL of Milli-Q water. The Petri dishes were sealed with parafilm and placed in a fridge in order to avoid evaporation of the EOs released from the NPs into the water. The water was changed after 24, 48, 72, 96, 120, 144, 168, 192, 216, 240, 264, 288, 312, 336, 360, and 384 h. The EO content was determined using reversed-phase HPLC as described above. In particular, 400 µL of the changed water was mixed with 600 µL MeOH, filtered with a 0.20 µm syringe filter, and injected into the HPLC system. Quantification of the released EOs was performed according to a calibration curve using the pure EOs. Cumulative release was determined for the different time intervals and related to the total amount of EOs released at the end of the experiment.

### 2.4. FTIR Imaging

FTIR mapping images of the spray-coated wood samples and a control wood sample without coating were acquired using a Jasco IRT-7000 Irtron infrared microscope coupled to a Jasco FTIR-4100 Fourier Transform Infrared microscope (Jasco Corporation, Easton, MD, USA). The analyses were performed in absorbance mode in the range of 4000–950 cm^−1^, with a 16× cassegrain objective, applying an aperture size of 50 × 50 μm, a mapping area of 25 × 25 points with an interval of 25 µm, a spectral resolution of 4 cm^−1^, and 16 scans. Raw FTIR spectra were smoothed using the Means-Movement method with a convolution width of 25 and baseline corrected between 2000 cm^−1^ and 950 cm^−1^. IR band areas around lignin-specific wavenumber maxima corresponding to aromatic C-H in plane deformation vibrations (1035 cm^−1^) and aromatic skeletal vibrations (1507 cm^−1^), as well as the PLGA-specific wavenumber maximum around 1080 cm^−1^ were utilized for the creation of FTIR imaging maps using Spectra Manager software (v. 2.15.01, Jasco Corporation, MD, USA).

### 2.5. Scanning Electron Microscopy

For SEM sample preparation, drops of the EOP suspensions were adsorbed onto a glass coverslip and air-dried at 25 °C. The cover slips were then attached to aluminum stubs using carbon tape and sputter-coated with gold in a Balzers MED 010 unit (Oerlikon Balzers, Balzers, Liechtenstein). Spray-coated wood samples and a native wood sample were cut to a size of ca. 5 × 5 mm, attached to aluminum stubs, and sputter-coated before SEM analysis. SEM analysis was conducted using a JSM 6010LA electron microscope (JEOL Ltd., Tokyo, Japan).

### 2.6. Confocal Laser Scanning Microscopy

One beech wood sample spray coated with EOL-TC was investigated using a Zeiss LSM 900 confocal laser scanning microscope equipped (Oberkochen, Germany)on Axio Observer Z1/7 inverted stand, with diode lasers with 405 nm, 488 nm, 561 nm, and 640 nm wavelengths, and using a plan-apochromat 20×/0.8NA objective (Carl Zeiss S.p.A., Milan, Italy). The measurements were conducted in a measurement area of 319.45 × 319.45 µm using fluorescence excitation for the specific illumination of lignin (553 nm) and carbohydrate (401 nm) moieties on the coated wood sample surface. The emission wavelengths were 568 nm for lignin and 422 nm for cellulose.

## 3. Results and Discussion

### 3.1. Characterization of Lignin Nanoparticles Loaded with Essential Oils (EOLs)

Table 1 provides a comprehensive breakdown of the two EOs utilized, including their detailed compositions and percentages. The four terpenoids—carvacrol, thymol, *p*-cymene, and γ-terpinene—represent the key compounds as they together account for around 80% of the two EOs. While EO-TC is dominated by carvacrol and thymol is present only in traces, the major components in EO-TV are thymol, *p*-cymene, and γ-terpinene.

The prepared EOLs had solid contents of 14.6 mg/mL (EOL-TC) and 15.5 mg/mL (EOL-TV), and EO contents of 5.1 mg/mL for EOL-TC and 4.3 mg/mL for EOL-TV (Table 2). The resulting drug-loading efficiencies (DLEs) were between 42 and 51% for the EOL preparations and between 32 and 39% for the EOP formulations, indicating a higher affinity of the EO compounds for lignin than for PLGA, even if the DLE for EOP-TV was almost as high as that for EOL-TV. The DLE as well as the drug-loading capacity (DLC) was higher for EOL-TC than for EOL-TV, while they were higher for EOP-TV than for EOP-TC, indicating different affinities of the carrier molecules lignin and PLGA towards the components of the respective EOs. This could be attributed to the higher content of hydroxylated compounds in EO-TC than in EO-TV (Table 1). Hydroxyl groups of the chemical compounds in the EOs favor interactions with the lignin carrier via hydrogen bonding in addition to π-stacking between aromatic rings. On the other hand, in the case of PLGA, no interactions via π-stacking can be expected and the EO entrapment is realized solely via hydrogen bonding. It was shown that carvacrol preferably exhibits hydrogen bonding between its OH group and a π-hydrogen of the aromatic ring of a second carvacrol while thymol tends to create hydrogen bonds between the OH groups of two thymol molecules. This tendency of thymol for classical hydrogen bonding could explain the higher affinity of EO-TV to PLGA nanoparticles rather than EO-TC, which contains mainly carvacrol [94]. It is known in the literature that the entrapment of EOs into PLGA NPs is generally low, which limits its use as a delivery system for EOs [42].

On average, the DLC for the EOLs were higher than for the EOPs, but the DLC of EOP-TV was in the same range as the one for EOL-TV. The results for the DLE and DLC of the two EOLs samples were lower than those reported in an earlier study [59], indicating that the EO concentrations determined there using an UV spectroscopy method were over-estimated compared with the EO concentrations determined by the more exact HPLC method applied in the present study. The DLE determined for the EOPs were comparable with results reported by Zhu et al. [95] where thymol was entrapped into PLGA microspheres and a maximum DLE of around 40% was achieved. However, the DLC obtained in the work of Zhu et al. [95] was significantly lower, with values slightly higher than 10%. It is known in the literature that the entrapment of EOs into PLGA NPs is generally low, which limits its use as a delivery system for EOs [42].

An SEM analysis of the aqueous EOP dispersions revealed particles with a smooth surface with a rather uniform size distribution, and size measurements resulted in particle diameters in the range from 100 to 700 nm (Figure 1). Interestingly, the EOPs did not show surface pores and hollow structures as was observed for the EOLs [59], an observation that was also reported elsewhere by Zhu et al. [95], while Moeini et al. [84] detected crystals of the loaded neem oil in the surface of the PLGA nanospheres. SEM images of the EOPs further revealed densely packed particles, indicating a certain tendency for agglomeration which was especially observed in the case of unloaded EOPs (Figure 1G–I). The incorporation of hydrophobic molecules like carvacrol, thymol, and *p*-cymene into the EOPs could contribute to an increased stability of the PLGA NPs against degradation by humidity and thus a lower tendency for agglomeration compared to unloaded PLGA NPs.

Based on the SEM images of the EOPs as well as SEM images of the EOLs published earlier [59], particle size measurements were conducted using Adobe Photoshop software (version 25.5.1). Interestingly, the EOPs showed much smaller average particle diameters than the EOLs and while unloaded PLGA NPs had a bigger average particle diameter than the EOPs, it was the opposite for unloaded LNPs (LNPs alone), which showed smaller average particle diameters than the loaded EOLs (Table 2). Further, the range of the observed particle diameters for the EOPs was much smaller than for the EOLs, which is evident from the respective standard deviations that were around 50–60% for the EOLs and only 25–30% for the EOPs. The respective size distributions of the lignin and PLGA NPs dispersions illustrated in Figure 2A further show a much broader distribution of the lignin NPs with a frequency of only 15–20% of the most abundant particle size fractions. The lignin NPs showed a tendency for additional relative maxima at 3200 and 4000 nm for EOL-TC, at 2600 nm and 4000 nm for EOL-TV, and at 2400 and 3400 for LNPs alone. In contrast, the respective frequencies of the most abundant particle size fraction for the PLGA NPs was around 60% and no particles with sizes above 730 nm were observed. The determined mean particle sizes and standard deviations for unloaded PLGA NPs (PLGA alone), EOP-TC, and EOP-TV are in good agreement with the literature values for PLGA NPs prepared using a dialysis method that were reported by Palocci et al. [89], but were lower than those reported for doxorubin-loaded PLGA NPs [81], even though the starting PLGA concentrations in the current study were significantly higher. Although the mean particle size of EOL-TV was only slightly lower than that of EOL-TC (Table 2), the respective particle size distributions in Figure 2A indicate a higher abundance of smaller NPs in EOL-TV. In EOL-TV, the particle size fractions with the highest frequency were between 600 nm and 1000 nm while the most abundant fractions in EOL-TC were between 1200 nm and 1800 nm.

### 3.2. Spray Coating of NP Dispersions onto Wood Samples

Beech wood samples were spray coated according to Figure 1 using the aqueous dispersions of the EOLs, EOPs, as well as of the pure EOs. The wood samples before and after spray coating with the EOLs and EOPs are illustrated in Figure 3(1). Spray coating with the EOLs resulted in brown-colored wood samples, as was observed in other studies where lignin was used in the form of LNPs for dip-coating [60] or as an additive for acrylic or polyurethane wood coatings [45,51]. On the other hand, the EOP coating did not affect the color of the beech wood samples. However, the surface of the samples apparently became rougher with the deposition of the EOP dispersion. Although the color change was quite evident for the EOL coatings, the respective coating uptake was rather low with 2.70% for EOL-TC and 2.54% for EOL-TV relative to the weight of the beech wood samples (Table 3). Interestingly, in the case of the PLGA nanoparticles, the coating uptake was lower and only about half that for the EOLs, indicating less affinity and less compatibility of PLGA nanoparticles with wood surfaces than lignin nanoparticles. However, the EOPs showed a strong tendency to agglomerate which caused precipitations of the EOPs in the nozzle of the nano-sprayer. This problem was confronted with a higher dilution of the EOP dispersions, but the agglomeration and subsequent precipitation could not be completely avoided, leading to a reduced coating uptake for the EOPs. In the case of EOL-TC and EOL-TV, the amounts of the EOs deposited on each sample was 19.1 mg and 13.4 mg, respectively, and therefore higher than in the case of the pure EO dispersions EO-TC alone (6.6 mg) and EO-TV alone (9.1 mg). This shows that the deposition of EOs on the beech wood samples was more efficient when applied in the form of EOLs entrapped into lignin NPs. The cause for this increased deposition in form of EOLs is assumed to be lignin’s ability to hinder EO evaporation, an effect that has been cited in the literature [69,77]. In contrast, the amounts of EO deposited, when applied as an EOP, were slightly lower than for the case of the EOLs as well as for the pure EO dispersions (Table 3).

The surfaces of the native and spray-coated wood samples were carefully investigated by SEM (Figure 4). Already at a low magnification of ×400, it is evident that the structure of the native wood (Figure 4(A1)) was almost completely covered by EOL-TC (Figure 4(B1)) and by EOL-TV (Figure 4(C1)). In the case of EOP-TC (Figure 4(D1)) and EOP-TV (Figure 4(E1)), the pattern of the native wood structure was more visible beneath the spray-coating layers than was observed for the lignin-based coatings.

At higher magnifications, the SEM photographs clearly show areas that were well coated by the respective NPs, but also a few areas where the native wood structure was still exposed. Interestingly, the PLGA nanoparticles appeared less defined than the lignin nanoparticles, and showed a tendency to fuse together, resulting in a film-like structure, which was especially evident in the case of EOP-TV (Figure 4(E2,E3)). However, under this kind of veil of fused EOP-TV, the shape of round NPs was still evident, indicating that this effect could be limited to the very surface of the EOP-TV spray coating. The fact that EOP-TC maintained some of its original spherical form in contrast to EOP-TV could be caused by a higher stability of EOP-TC. This could be conferred by the incorporation as well as superficial attachment of carvacrol, the main component in EO-TC (Table 1), which is considered a more hydrophobic molecule than thymol, thus providing higher protection for PLGA against hydrolysis by water molecules compared to EO-TV. Due to the film-like appearance of the EOP-TV spray coating, it was not possible to determine the particle sizes for that sample. Besides the loss of their perfect spherical shape, which was observed by SEM for PLGA NPs in aqueous dispersions, EOP-TC showed larger particle sizes on the beech wood surfaces after application via spray coating (Figure 2B). The average particle size increased by 60% (370 vs. 594 nm) and the standard deviation almost doubled (125 vs. 207 nm), which could be caused by the agglomeration of primary NPs. Agglomeration was also indicated by the particle size fractions, which ranged from 800 nm to 1400 nm, which were not observed for EOP-TC when they were investigated in aqueous dispersion. In general, PLGA is characterized by a low glass transition temperature around 38 °C that could be even lower in conditions under pressure such as when they are passing through the nozzle of the nano-sprayer [83]. It may be expected that during spray coating, the morphology of the EOPs was affected which leads to agglomerated clusters in the case of EOP-TC (Figure 4(D3)), and to a film-like coating appearance in the case of EOP-TV (Figure 4(E3)). The difference in the appearance of EOP-TC and EOP-TV could be caused by the respective loadings, as it was reported in the literature that the loading can affect the swelling and degradation of PLGA [42], and EO-TC apparently conferred a certain stability increase to the PLGA carrier matrix. For the LNPs, an effect of the loadings on LNP stability was not observed in this study. PLGA is a polymer that tends to agglomerate and is hydrolyzable by water, which was used as the co-solvent in the nanoparticle preparation protocol. Even if the EOP dispersions were freshly prepared and stored in the fridge before using them for spray coating, degradation of PLGA nanoparticles cannot be completely excluded during the spray coating application. This could also explain why some areas of the wood samples’ surface were not covered and why the EOP-TV-coated wood sample appeared to be covered with a film-like coating.

Larger particle sizes were also observed for EOL-TC and EOL-TV when the spray-coated beech wood sample surfaces were analyzed by SEM, but the shape of the EOLs was still perfectly round and no agglomeration effects were evident as in the case of PLGA. The mean particle size increased by 26% for EOL-TC (1519 vs. 1912 nm), while it increased only by 9% for EOL-TV (1414 vs. 1535 nm) when comparing the spray-coated particles with those in the aqueous dispersions. The tendency for larger particle sizes was also evident from the particle size distributions illustrated in Figure 2A,B, which show higher frequencies for the particle size fractions >2400 nm for both EOL-TC and EOL-TV. However, the stronger increase in the mean particle size for EOL-TC compared to EOL-TV when the spray-coated NPs were measured could indicate a higher tendency of low-particle-size fractions of EOL-TC to fill wood pores and assemble on the very surface of the wood samples, leaving NPs with larger particle sizes in the top layers of the spray coatings. Another possible explanation could be a higher dispersion of the smallest EOL-TC particles, causing them not to deposit on the beech wood samples. A similarly wide particle size distribution of lignin NPs was presented in the work of Cusola et al. [96], where a self-standing membrane based on mainly lignin NPs (92%) and cellulose nanofibrils as structuring additives were produced. SEM images of the cross-sections of the membranes showed an agglomerate of lignin NPs with sizes of several micrometers but also particles smaller than 1 µm. 

The beech wood sample surfaces of the EOL-treated wood showed fewer open vessels and ray cells, which were instead more evident in the SEM images of the native wood (Figure 4(A1,A2)) and the EOP-coated wood samples (Figure 4(D1,E1)). The wider size distribution and larger particle diameter of the lignin NPs apparently positively influenced the coating distribution over the wood sample surface, filling the large wood vessels, which in beech wood may have sizes of 50 µm and more. The fact that the EOLs precipitated into the vessel lumen is especially evident in Figure 4(C2) where the walls of the vessel filled with EOL-TV can be clearly distinguished.

### 3.3. Release Experiments of Essential Oils from the Spray Coatings

The spray-coated beech wood samples were immersed in water for a total of 18 days. The water was changed every 24 h and analyzed by HPLC to quantify the EOs released into the water. The EOL-coated wood samples lost part of their lignin coating during the release experiment, resulting in a lighter coloration, while the removal of the EOP coating was evident by the surface structure change in the wood samples, as illustrated in Figure 3(2). Although the EOL-coated samples became much lighter in color, the samples after the release experiment evidently retained a certain amount of lignin coating. The EOP-coated samples showed a kind of rough or brittle-like surface, which was apparently removed after the release experiment. Interestingly, the signs of a microbiological attack in the form of black dots and a dark halo were registered on the surface of the sample spray-coated with EOP-TC (Figure 3(2F), yellow arrows).

Figure 5 shows the cumulative EO release curves of the wood samples spray-coated with the EOLs and EOPs, as well as the wood samples that were sprayed with aqueous dispersions of the pure EOs. Comparing the release of the three different application forms of the EOs, it becomes evident that on day 9 of the leaching experiment, the complete release of the EOs has already been achieved for the wood samples treated with the pure EO dispersions (Figure 5). Only in the case of the EOLs and EOPs, there was still EO present on the wood samples. Meanwhile, at that point, around 10–15% of the original EO loading was still present on the wood samples treated with the EOL dispersions, and the residual EO loading on the wood samples treated with the EOP dispersions was only 1–2%. In the first 24 h, the release was comparable for the three different application forms for both TC and TV, where around 45–50% of the incorporated EO-TC and 33–43% of the EO-TV were released. This initial burst release can be attributed to EO loaded at the surface of the nanosphere, which was released without retention by the carrier matrix. This effect was reported before for different cases and different carrier materials, such as lignin [39,59,97] as well as PLGA [84]. However, after 24 h in the case of TC and after 48 h in the case of TV, the cumulative release was significantly lower for EOL-TC and EOL-TV and their release rate decreased faster compared to the loaded PLGA nanoparticles and the dispersions of the pure EOs. This indicates a stronger interaction between the EO compounds, such as carvacrol, thymol, and *p*-cymene, and the lignin matrix leading to increased retention of the EOs by the lignin nanoparticles. Interestingly, the difference in the release rates of EOL-TC and EOL-TV was much less pronounced compared to the results obtained in a previous work where the release rates were determined for the same lignin NPs but dispersed in an aqueous medium [59]. This could mean that when deposited on wood, a loading-dependent release rate was attenuated by the type of substrate and thus the release curves for EOL-TV and EOL-TC were almost identical.

After the release experiments, the beech wood samples were dried and analyzed using SEM in order to compare their surface morphology with that of the spray-coated samples before their immersion in water for the determination of the release of the EOs (Figure 6). Compared to the SEM images of the spray-coated samples (Figure 4), the number of lignin nanoparticles (Figure 6(A1–A3,B1–B3)) as well as PLGA NPs (Figure 6(C1–C3,D1–D3)) detected was lower and the wood surface structure was more evident. In particular, open ray cells were observed after the release experiment both in the case of the EOPs (Figure 6C1, blue circle) and in the case of the EOLs (Figure 6(B1,B2), blue circle), and these structures were partly filled with EOLs. On the surface of the beech wood sample spray coated with EOP-TC (Figure 6(C1–C3)), fungal hyphae were detected (yellow arrows) which might indicate that the amount of EOs taken up by the spray coating of EOP-TC was not sufficient to prevent fungal contamination. Indications for that were also found in the photos of the respective beech wood samples illustrated in Figure 3(2) and mentioned above.

The particle sizes of the NPs that remained on the beech wood samples after the leaching experiment were smaller for the EOLs compared to before the wood samples were immersed into water, and their particle size distribution was significantly narrower (Figure 2C). This decrease was more pronounced for EOL-TV than for EOL-TC and the mean particle size of EOL-TV after the release experiment was as low as those determined for the PLGA NPs (Table 2). Considering the particle size distributions of EOL-TC and EOL-TV after the release experiment, it can be concluded that the particle size of EOL-TV was considerably smaller than that of EOL-TC, as was already indicated by the respective results from the aqueous dispersions and the spray-coated beech wood samples (Figure 2A,B). The fact that the mean particle size after leaching was significantly lower for both EOL-TC and EOL-TV indicates that certain particle-size-dependent dynamics occurred during the spray coating of the beech wood samples. Thus, it is assumed that, after the smallest particle size fractions had filled the surface pores, the next larger particle size fractions created the first coating layer, which was then covered by the biggest particle size fractions in order to create the final coatings illustrated in (Figure 4(A1–A3,B1–B3)). Similarly, Cusola et al. [98] observed a structured morphology of pure lignin NP coatings that they prepared by casting on solid silica supports. However, their findings were the opposite of the data presented here, with the smallest particles on the surface and the largest particles on the bottom of the coating layers. Most likely, the production method has a significant effect on the structure of the packed particles in the lignin NP coating, leading to different results for solvent casting and for spray coating.

In the case of the PLGA NPs, the particle size distribution did not change significantly after the release experiment. However, in the case of EOP-TV, the film-like appearance of the spray coating illustrated in Figure 4(E3) was almost gone on the investigated surface of the sample after leaching, and it was therefore possible to measure the particle size of EOP-TV (Table 2, Figure 2C). This indicates that the EOP-TV spray coating exhibited that film-like appearance only on the very surface of the coating and deeper layers were still present as distinct but agglomerated PLGA NPs, as pointed out above. The fact that after the release experiment, the PLGA NPs in both cases EOP-TC and EOP-TV, respectively, were still detected, confirms a certain stability of the prepared PLGA NPs even after full immersion in water for a period of 19 days, even though PLGA generally has a short degradation period [91]. Extended stability of PLGA was reported by Moroishi et al. [99], who prepared thin and flexible sheets by spin-coating PLA/PLGA blends for the controlled release of hydrophilic compounds and performed release experiments in PBS buffer for as long as 100 days.

### 3.4. FTIR Imaging

The surfaces of the spray-coated wood samples were investigated by FTIR mapping in order to evaluate the homogeneity of the produced coatings. Uncoated wood (Figure 7A,B,K) was compared with wood samples spray coated with EOL-TC (Figure 7C,D) and EOL-TV (Figure 7G,H). The FITR images were mapped at a wavenumber of 1135 cm^−1^ (Figure 7A,C,E,G,I), which represents aromatic C-H in-plane deformation vibration modes specific for lignin, and at 1507 cm^−1^ (Figure 7B,D,F,H,J), which represents lignin aromatic skeletal vibrations. The FTIR absorbance mappings of the spray-coated wood samples show a higher absorbance in almost the whole investigated area, although a small area with low absorbance values in the uncoated control samples can be seen on the EOL-coated wood samples (Figure 7C,D,G,H). At the second investigated wavenumber of 1507 cm^−1^, the sample coated with EOL-TC similarly showed low absorbance areas (Figure 7D), while those areas were almost absent on the sample coated with EOL-TV (Figure 7H), indicating a more successful coating when the EOL-TV dispersion was used. 

The FTIR spectra of the wood samples coated with the EOPs were instead mapped at a wavenumber maximum of around 1080 cm^−1^, which was identified as a specific peak for PLGA (Figure 7L–O). A comparison of the measurement areas of the samples coated with EOP-TC (Figure 7L) and EOP-TV (Figure 7N) indicated a less homogeneous coating of the measurement area of the latter with larger spots of low absorbance where a less successful deposition of EOP-TV was evident. A reduced homogeneity of the sprayed coatings, especially in the case of the PLGA nanoparticles was also observed by the SEM analysis, as pointed out above (Figure 4).

When comparing the FTIR mappings of the beech wood samples spray coated with EOL-TC before (Figure 7C,D) and after the release experiment (Figure 7E,F), the overall absorbance of the mapped area was lower after leaching, although there were areas with similarly high absorbance values as before. The same was the case for the investigated beech wood sampled spray-coated with EOL-TV. Similarly, the wood samples spray-coated with EOP-TC (Figure 7L) and EOP-TV (Figure 7N) show lower absorbance values after the release experiment (Figure 7M,O). However, the absorbance of the wood samples after leaching was higher than that of the untreated wood in all cases, confirming the findings from the SEM analysis, where a certain abundance of lignin and PLGA NPs was still detected on the wood samples after leaching (Figure 6).

### 3.5. Confocal Laser Scanning Microscopy

One beech wood sample coated with EOL-TC was investigated by fluorescence microscopy using a Zeiss LSM900 laser-scanning microscope. The resulting three-dimensional image, which was constructed by the Zeiss ZEN software (version 3.8) based on a scan of 48 layers in the z direction with a step size of 1 µm after applying the respective fluorescence filters for lignin and cellulose, is illustrated in Figure 8A. For lignin, a fluorescence excitation wavelength of 553 nm was used, and 401 nm was used for cellulose. The respective channel-specific images as well as the overlay are displayed in Figure 8B (red channel—lignin; green channel—cellulose). In Figure 8C, seven layers in the z direction with a 3 µm step size are displayed, showing the deposition of red fluorescent LNPs on the very surface of the spray-coated beech wood samples. In deeper surface layers, green fluorescent cellulose fibers were more evident, with deposited LNPs around them filling the empty spaces of the beech wood samples’ surface structure.

## 4. Conclusions

Lignin nanoparticles appeared as larger-diameter particles with a broader particle size distribution than PLGA nanoparticles, and showed a higher efficiency and capacity to entrap EOs of TC and TV. EO-TC was more efficiently entrapped into lignin NPs than EO-TV, most likely because of a higher content of hydroxylated compounds in EO-TC, leading to increased hydrogen bonding between the EO and lignin carrier matrix. 

The SEM analysis of the lignin coatings showed that they formed a rather uniform layer that covered most of the surface pores of the spray-coated beech wood samples, such as open vessels and ray cells. The PLGA NPs had irregular shapes and they formed agglomerated clusters after spray coating. The PLGA coatings had a film-like appearance and the presence of wood structural compartments like parenchymatic cells in the SEM images indicated a less effective surface coating. 

The particle sizes of EOL-TC were significantly higher than those determined for EOL-TV, and for both, the mean particle size of the spray coatings was higher than for their aqueous dispersions. The reason for this could be that the smallest NPs collected in the deeper layers of the prepared spray coatings and filled smaller cell compartments like parenchymatic cells, and eventually layers of larger NPs covered these base layers. 

The PLGA NPs had the tendency to agglomerate after spray coating on wood, which resulted in clusters with average particle diameters that were almost double compared to the aqueous dispersions. The PLGA NPs with incorporated EO-TV showed a film-like appearance that disappeared after the release experiment. 

The release experiments showed that were was a considerable delay when EO-TC was applied as EOL-TC, while application as EOP-TC did not significantly slow down the leaching of the loaded EO. In the case of EO-TV, both the PLGA and lignin matrices showed a strong delay effect on the release of the incorporated EO, but lignin outperformed PLGA for both applied EOs as a polymeric carrier matrix for a biocidal delivery system containing EOs.

## Figures and Tables

**Figure 1 polymers-16-00947-f001:**
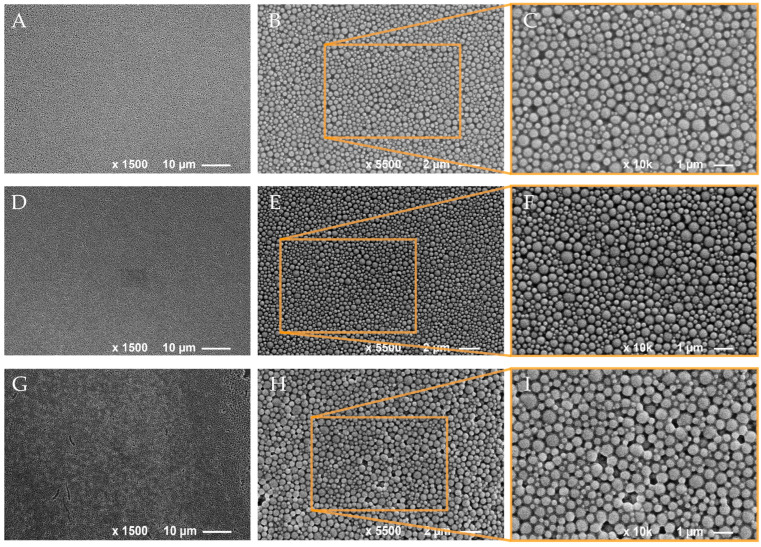
SEM images of PLGA nanoparticles containing essential oils from *T. capitatus* (**A**–**C**) and *T. vulgaris* (**D**–**F**), as well as unloaded PLGA nanoparticles (**G**–**I**).

**Figure 2 polymers-16-00947-f002:**
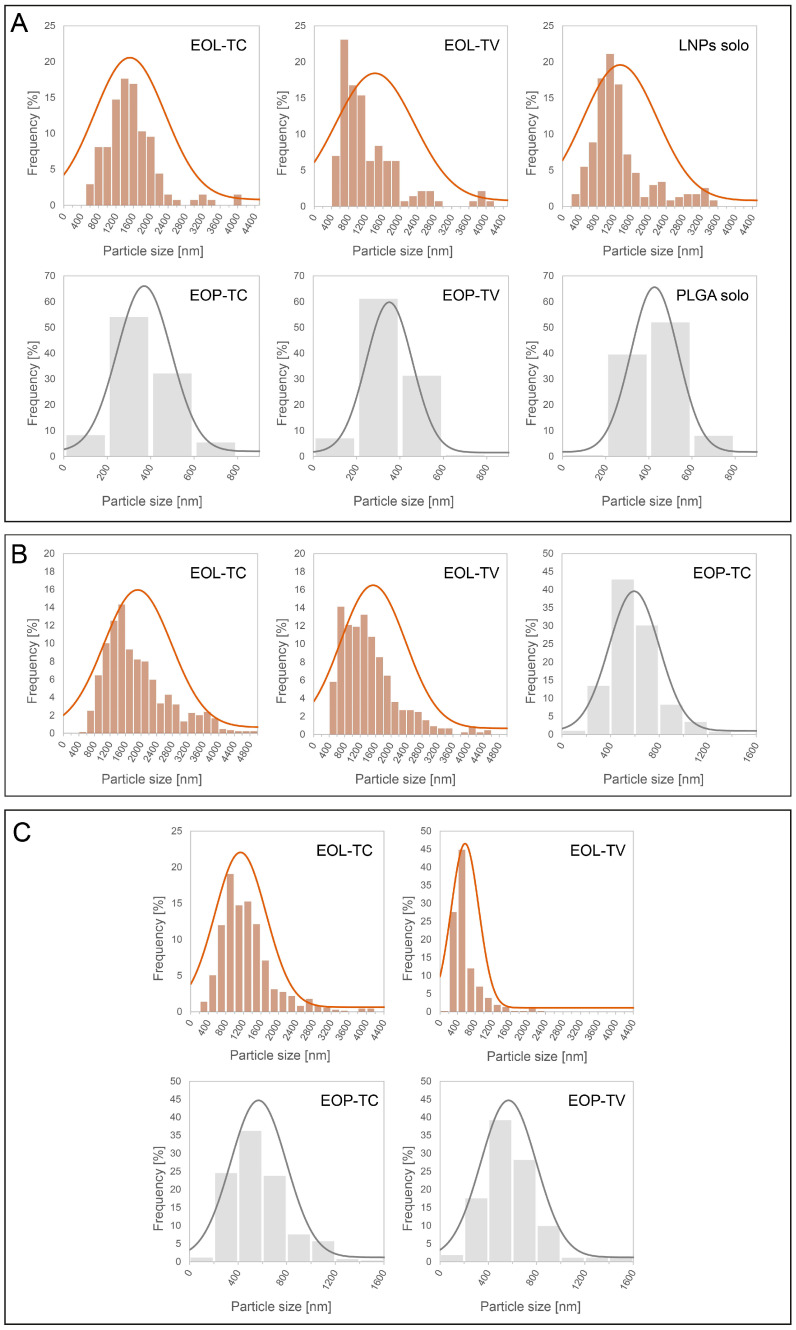
Size distributions of lignin and PLGA nanoparticles loaded with essential oils from *T. capitatus* (EOL-TC, EOP-TC) and *T. vulgaris* (EOL-TV, EOP-TV), and unloaded lignin and PLGA nanoparticles (LNPs alone, PLGA alone) in aqueous dispersions (**A**), on the beech wood samples surfaces after spray coating (**B**), and on the beach wood samples surfaces after the release experiment (**C**).

**Figure 3 polymers-16-00947-f003:**
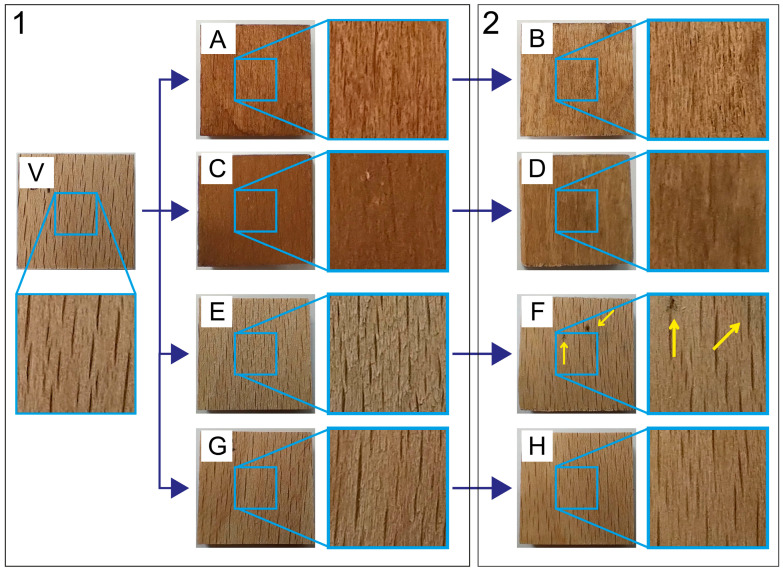
A virgin beech wood sample before spray coating (V) and the beech wood samples spray coated with aqueous dispersions of lignin nanoparticles loaded with essential oils from *T. capitatus* (EOL-TC, **A**,**B**) and from *T. vulgaris* (EOL-TV, **C**,**D**), and PLGA nanoparticles loaded with essential oils from *T. capitatus* (EOP-TC, **E**,**F**) and from *T. vulgaris* (EOP-TV, **G**,**H**) before (**1**) and after the release experiment (**2**).

**Figure 4 polymers-16-00947-f004:**
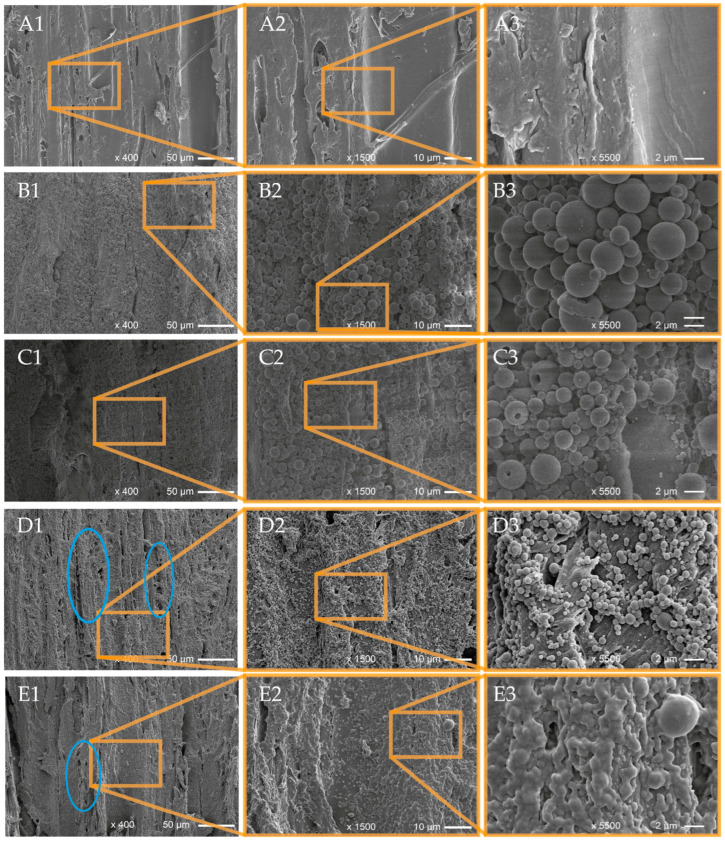
SEM images of native uncoated wood samples cut in tangential section (**A1**–**A3**), and wood samples coated with EOL-TC (**B1**–**B3**), EOL-TV (**C1**–**C3**), EOP-TC (**D1**–**D3**), and EOP-TV (**E1**–**E3**).

**Figure 5 polymers-16-00947-f005:**
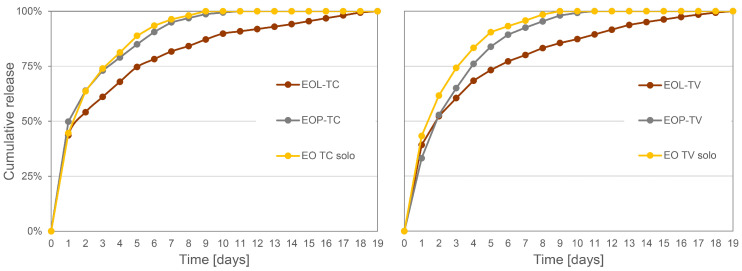
Cumulative release of essential oils from beech wood samples spray coated with lignin nanoparticles containing essential oils from *T. capitatus* (EOL-TC) and *T. vulgaris* (EOL-TV); PLGA nanoparticles containing essential oils from *T. capitatus* (EOP-TC) and *T. vulgaris* (EOP-TV); as well as dispersions of the pure essential oils (EO-TC alone, EO-TV alone).

**Figure 6 polymers-16-00947-f006:**
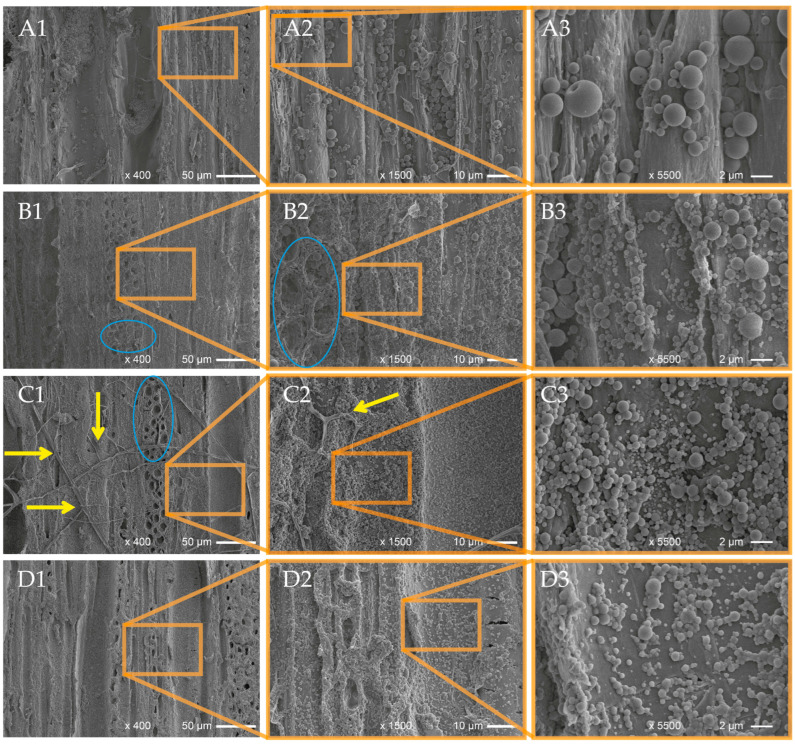
SEM images of beech wood samples coated with lignin nanoparticles loaded with essential oils from *T. capitatus* (EOL-TC, **A1**–**A3**) and from *T. vulgaris* (EOL-TV, **B1**–**B3**), and wood samples coated with PLGA nanoparticles loaded with essential oils from *T. capitatus* (EOP-TC, **C1**–**C3**) and from *T. vulgaris* (EOP-TV, **D1**–**D3**) after the release experiment.

**Figure 7 polymers-16-00947-f007:**
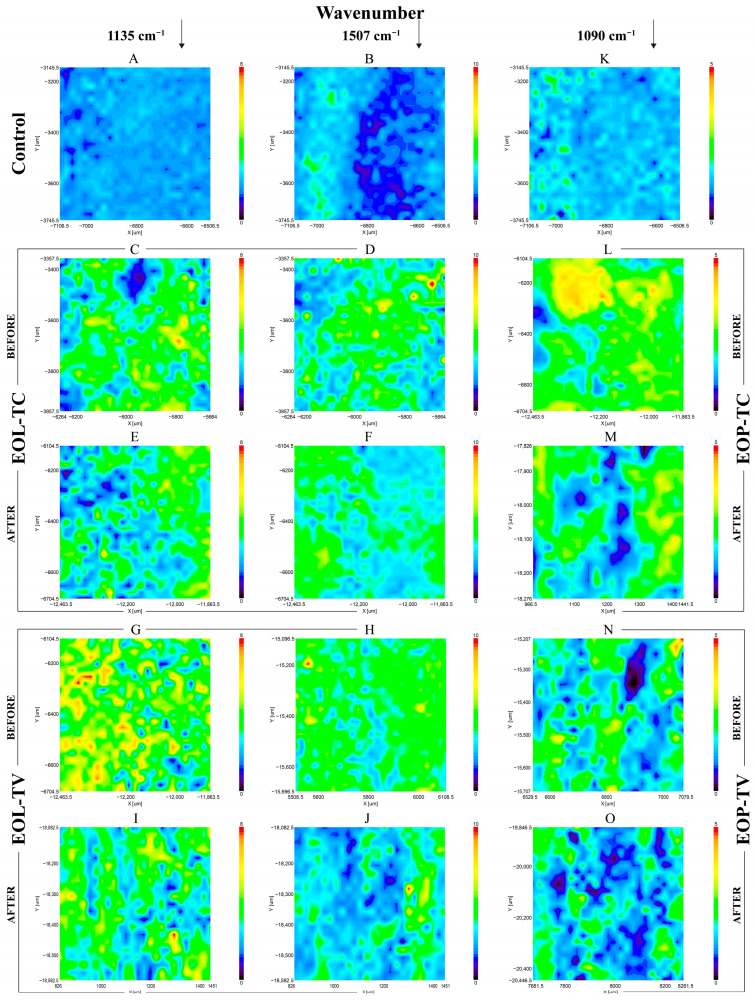
FTIR mapping images of native wood (**A**,**B**,**K**); a wood sample spray-coated with EOL-TC before (**C**,**D**) and after (**E**,**F**) the leaching experiment; a wood sample coated with EOL-TV before (**G**,**H**) and after (**I**,**J**) leaching; a wood sample coated with EOP-TC before (**L**) and after (**M**) leaching; and a wood sample coated with EOP-TV before (**N**) and after (**O**) leaching. The left column was mapped at a wavenumber maximum of 1135 cm^−1^, the center column at 1507 cm^−1^, and the right column at 1090 cm^−1^.

**Figure 8 polymers-16-00947-f008:**
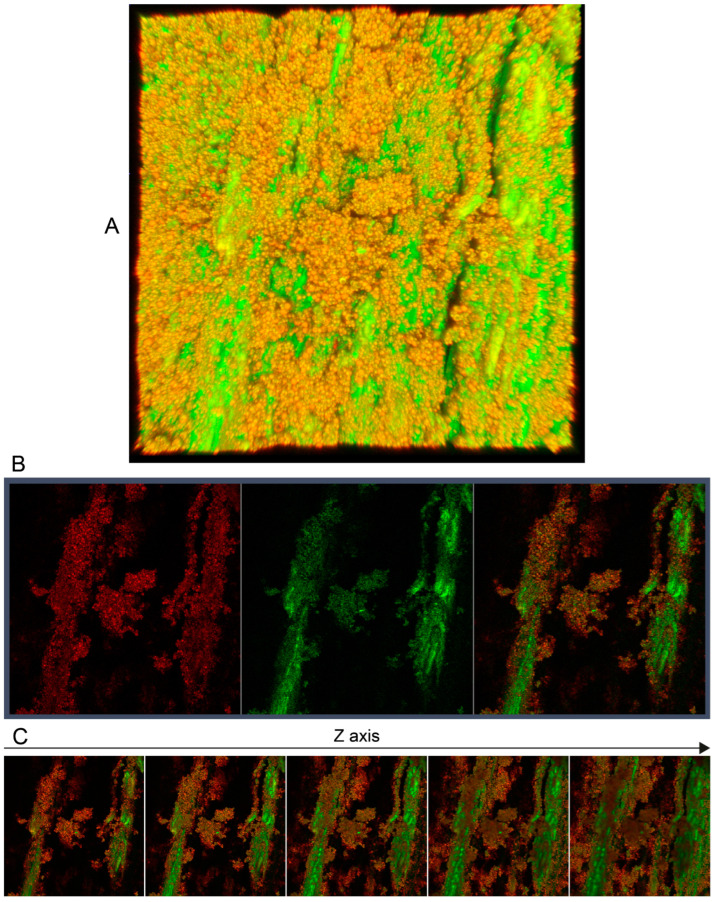
(**A**) Three-dimensional image of the beech wood sample surface spray-coated with EOL-TC dispersion obtained using a Zeiss LSM900 laser-scanning microscope using lignin- and cellulose-specific fluorescence filters. (**B**) Images of the same layer in the z direction after application of a lignin-specific (left), a cellulose-specific (center), and both (right) fluorescence filters of the spray-coated beech wood sample surface. (**C**) Different z layers in 3 µm steps of the spray-coated beech wood sample surface after application of both fluorescence filters.

**Table 1 polymers-16-00947-t001:** Main chemical compounds contained in the essential oils of *Thymus capitatus* (EO-TC) and *T. vulgaris* (EO-TV).

EO-TC		EO-TV	
Component	%	Component	%
carvacrol	68.6	thymol	47.9
*p*-cymene	7.7	*p*-cymene	15.8
γ-terpinene	6.8	γ-terpinene	10.0
β-caryophyllene	2.6	carvacrol	4.4
β-myrcene	1.8	linalool	4.1
linalool	1.5	β-caryophyllene	2.1
α-thujene	1.2	β-myrcene	2.0
α-terpinene	1.1	borneol	1.3
α-pinene	0.9	α-terpinene	1.3
terpinene 4-ol	0.7	α-thujene	1.2
thymol	0.6	camphene	1.1

**Table 2 polymers-16-00947-t002:** Solids content, essential oil (EO) content, drug-loading efficiency (DLE), and drug-loading capacity (DLC) of unloaded lignin NPs (LNP alone), LNPs with entrapped EOs from *T. capitatus* (EOL-TC) and *T. vulgaris* (EOL-TV), unloaded PLGA NPs (alone), and PLGA NPs loaded with essential oil from *T. capitatus* (EOP-TC) and from *T. vulgaris* (EOP-TV).

Sample	Solids [mg/mL]	EO [mg/mL]	DLE [%]	DLC [%]	Particle Size [nm]	Particle Size after Spray Coating [nm]	Particle Size after Release Experiment [nm]
LNP alone	16.7	-	-	-	1325 ± 680	n. d.	n. d.
EOL-TC	14.6	5.1	51	35	1519 ± 814	1912 ± 870	1273 ± 608
EOL-TV	15.5	4.3	42	28	1414 ± 913	1535 ± 841	571 ± 313
PLGA NPs alone	11.4	-	-	-	426 ± 109	n. d.	n. d.
EOP-TC	15.4	3.3	32	22	370 ± 125	594 ± 207	568 ± 229
EOP-TV	13.8	4.0	39	29	351 ± 107	n. d.	581 ± 222

**Table 3 polymers-16-00947-t003:** Coating uptake per beech wood sample spray coated with aqueous dispersions of lignin nanoparticles loaded with essential oil from *T. capitatus* (EOL-TC) and from *T. vulgaris* (EOL-TV, C and D); PLGA nanoparticles loaded with essential oil from *T. capitatus* (EOP-TC, E and F) and from *T. vulgaris* (EOP-TV); and the respective pure essential oils (EO-TC alone, EO-TC alone).

Sample	Coating Uptake per Wood Sample		EO Uptake
	[mg]	[%]	[mg]
EOL-TC	54.7	2.70	19.1
EOL-TV	48.2	2.54	13.4
EOP-TC	24.1	1.30	5.2
EOP-TV	28.7	1.44	8.3
EO-TC alone	6.6	0.36	6.6
EO-TV alone	9.1	0.50	9.1

## Data Availability

Data are contained within the article.

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
