# Peer review of "Spray Coating of Wood with Nanoparticles from Lignin and Polylactic Glycolic Acid Loaded with Thyme Essential Oils"

_polymers, 2024, doi:10.3390/polym16070947_

Round 1

Reviewer 1 Report

Comments and Suggestions for Authors

In this manuscript, the author reported spray-coating of wood with bio-nanoparticles, which is a meaningful work. In addition, the whole manuscript was in a good organizing and writing, the following issues should be addressed before acceptance:

(1) About 15 articles from the corresponding authors group were cited, which is too much.

(2) Section 3 should be Results and Discussion

(3) As multiple component exist in the thyme essential oils, for the cumulative release, how can you determined its content within the nanoparticles?

(4) Data results should summarized with Conclusion.

Comments on the Quality of English Language

Moderate editing of English language required

Author Response

(1) About 15 articles from the corresponding author’s group were cited, which is too much.

We removed three articles from the corresponding author`s group, and the result is a 13% autocitation rate, which is below the threshold of 14% given by the Section Managing Editor. We wish that this can be accepted by the Reviewer.

(2) Section 3 should be “Results and Discussion”

Thanks for bringing it to attention. We changed the heading of section 3 accordingly.

(3) As multiple component exist in the thyme essential oils, for the cumulative release, how can you determined its content within the nanoparticles?

The quantities of the released essential oil were determined in the aqueous medium where the beech wood samples were immersed in using HPLC. The quantification was done using a calibration curve with different concentrations of the pure essential oils.

(4) Data results should summarized with Conclusion.

We tried to summarize the results of the different analyses in the Conclusions section as synthetic as possible in single paragraphs (page 22, lines 559-584). However, if the reviewer considers this incomplete, we can expand the descriptions of the results in the Conclusions section accordingly.

Moderate editing of English language required

We reviewed again the text thanks to a collegue who is mother tongue

Reviewer 2 Report

Comments and Suggestions for Authors

Dear Editors, Dear Authors,

The manuscript presents an experimental study on spray-coating of wood with nanoparticles from lignin and polylactic glycolic acid loaded with thyme essential oils. The work is very interesting from a practical point of view and it is written very well. The researchers employed several research techniques such as: Scanning Electron Microscopy (SEM), Fourier – Transform Infrared Spectroscopy (FTIR) imaging, a release study with High - Performance Liquid Chromatography (HPLC) as well as Confocal Laser Scanning Microscopy. The method used in the research is clear and understandable. Undoubtedly, the authors master the approach well. However, below are questions and remarks that should be corrected before the manuscript could be consider for publication in Polymers.

  1. My question is connected to the nebulizer, that was attached to the chamber wall. I am afraid that more particles reached the piece of wood closer to the nebulizer (on the left side on Scheme 1) than the piece further away (on the right side on Scheme 1). The coatings obtained in this way on these pieces of wood are not the same, e.g. they do not have the same thickness.
  2. Paragraph 2.2. Release experiments: How many repetitions were performed for each release? The absence of error bars in Figure 5 suggests that only 1 test was performed. It would be better to do at least 3 repetitions.

3.      Paragraph 2.2. Release experiments: Was the acceptor fluid stirred during the release study? At what temperature was the release carried out? Storing the preparations in the fridge suggests that the temperature was approximately 6°C. If the proposed preparation is to be widely used, the release at ambient temperature would be suggested.

  1. Figure 5: x axis label - it should be time and days in bracket [].
  2. Paragraph 3.5. lines 538-541 – there are information that should be in Methods, in paragraph 2.6. i.e. the name of the microscope, software, the number of scans layers etc.
  3. Line 210; it should be bracket added, (EOL-TV).

Author Response

Dear reviewer many thanks for your comment below our answers.

My question is connected to the nebulizer, that was attached to the chamber wall. I am afraid that more particles reached the piece of wood closer to the nebulizer (on the left side on Scheme 1) than the piece further away (on the right side on Scheme 1). The coatings obtained in this way on these pieces of wood are not the same, e.g. they do not have the same thickness.

The coating setup and the position of the nebulizer was carefully studied in pre-tests in order to assure that the deposition of the NPs on the single wood samples was uniform. The authors can understand the considerations of the reviewer, but we can also assure that the NPs deposition was uniform based on the standard deviation of the respective coating uptake, which was lower than 5% for all cases. 

  1. Paragraph 2.2. Release experiments: How many repetitions were performed for each release? The absence of error bars in Figure 5 suggests that only 1 test was performed. It would be better to do at least 3 repetitions.

      In order to assess the repeatability of the release experiment, a set of two experiments           was run in parallel using a sample of EOL-TC. The error of the method resulted as low as 1.6% at the most, and therefore the complete set up of the release experiment, including the samples of EOL-TC, EOL-TV, EOP-TC, EOP-TV, EO TC solo, and EO TV solo, was run in single tests after the robustness of the method was confirmed. We agree that in theory a triple determination would be the preferred approach. However, the time and resource request of the spray-coating method using a prototype and using six different formulations of NPs was considered too high for a “proof of concept” study, which will for sure need further experiments during the development of the spray-coating prototype towards a wider and eventually commercial use. The further experiments in the planning phase are focused on smaller aspects of the overall concept allowing tests in triple determination.

  1. Paragraph 2.2. Release experiments: Was the acceptor fluid stirred during the release study? At what temperature was the release carried out? Storing the preparations in the fridge suggests that the temperature was approximately 6°C. If the proposed preparation is to be widely used, the release at ambient temperature would be suggested.

The acceptor fluid was not stirred during the release study. The intention was to simulate a release experiment of “ultra-high” (100%) humidity. The temperature was approx. 6°C. In this first test of the wood conservation, the fridge set-up was used in order to avoid evaporation of the EOs. Evaporation would have compromised the quantification of the released EOs. Therefore, the setup was chosen accordingly. We are aware that there are further steps to implicate in order to develop the preparation to a level when it can be widely applied and the respective experiments are currently in a planning phase.

  1. Figure 5: x axis label - it should be time and days in bracket [].

The figure was corrected accordingly.

  1. Paragraph 3.5. lines 538-541 – there are information that should be in Methods, in paragraph 2.6. i.e. the name of the microscope, software, the number of scans layers etc.

This information is also reported in more detail in paragraph 2.6. However, we considered it important to report some of this this information also in paragraph 2.6, where the figures are presented. If the Reviewer considers it as redundant, we can cancel these lines from paragraph 2.6.

  1. Line 210; it should be bracket added, (EOL-TV).

Thanks for the attention, we corrected accordingly.

Reviewer 3 Report

Comments and Suggestions for Authors

In this work, authors demonstrated the application of lignin nanoparticle and PLGA nanoparticle as carriers to load essential oils, followed by spray coating onto the wood for wood preservation purpose. To tell the EO loading and release, the loading efficiency, loading capacity, release rate, as well as particle size, wood surface morphology, and surface mapping before and after spray coating were illustrated.  The experimental observations were well described, and the results were relatively well explained. Supplements are required in a few spots to enhance the reasoning and address the confusion. In terms of the experimental design, more clarification and justifications are needed to show the reason for the design. Therefore, this paper is suggested to have major revisions before approval for publication. Detailed comments are listed as follows:

1.      On page 3, section 2.2, the spray coated wood samples were dried at 40 C for 2 hours. What is the moisture level and/or water activity in spray coated wood after 2 hours drying? High moisture promotes bacteria and fungi growth, which is against the purpose of preservation. So, evidence to ensure sufficient moisture removal after 2 hours of drying is necessary and important.

 2.      On page 13, section 3.3, part of lignin and PLGA coating were lost during the release experiment. What is the affinity between lignin/PLGA and wood surface and what contributes to the adhesion? Can lignin/PLGA provide good enough adhesion to the wood to be qualified as coating carrier material for the objective of this work? 

3.      On page 14, many large particles were observed to be washed away during leaching experiments, which was explained by the particle-size -dependent dynamics. Is this deposit dynamics related to the horizontal spray set up (spray pattern is parallel to the wood) as shown in Scheme 1? Why not spray directly (vertically) towards the wood to increase the z direction spray uniformity and potentially increase coating uptake? 

4.      On page 6, line 255, the EO loaded EOPs are smaller in size than the unloaded PLGA NPs. What is the reason for this decreased size and how to prove the encapsulation of EO in PLGA NPs in this case?   

5.      On page 9, lines 307-309, EOs are deposited at a higher volume when loaded in NPs as compared to EO-solo. What is the possible explanation? 

6.      The preparation and spray coating method of EO TC/TV-solo is missing in the materials and methods section.

Author Response

  1. On page 3, section 2.2, the spray coated wood samples were dried at 40 C for 2 hours. What is the moisture level and/or water activity in spray coated wood after 2 hours drying? High moisture promotes bacteria and fungi growth, which is against the purpose of preservation. So, evidence to ensure sufficient moisture removal after 2 hours of drying is necessary and important.

The moisture level of the coated wood samples after 2 hours drying was around 5% relative to the dry weights of the samples, which was lower than the humidity determined for the beech wood samples under standard conditions at room temperature (about 6-7%).

  1. On page 13, section 3.3, part of lignin and PLGA coating were lost during the release experiment. What is the affinity between lignin/PLGA and wood surface and what contributes to the adhesion? Can lignin/PLGA provide good enough adhesion to the wood to be qualified as coating carrier material for the objective of this work? 

We assume that the affinity between lignin and PLGA NPs to the wood surface are based on hydrogen bonding between hydroxyl groups of lignin and PLGA NPs and hydroxyl groups of the wood constituents such as cellulose, hemicelluloses and lignin. A second assumed mode of action is a sterical entrapment of the NPs within the surface layers of the wood structure, as observed by SEM analysis and stated in the Results and Discussion section (page 13: lines 389-396; page 16: lines 451-452).

  1. On page 14, many large particles were observed to be washed away during leaching experiments, which was explained by the particle-size -dependent dynamics. Is this deposit dynamics related to the horizontal spray set up (spray pattern is parallel to the wood) as shown in Scheme 1? Why not spray directly (vertically) towards the wood to increase the z direction spray uniformity and potentially increase coating uptake? 

At this point of our studies, we assume that the deposit dynamics are dependent on the spray setup. However, more studies are needed to confirm this. For future investigations, different spray set-ups are planned while the prototype will be further developed. One of the approaches will definitely be a set-up where they spray is mounted on the cover of the Plexiglas box and spraying will be conducted towards the wood.

  1. On page 6, line 255, the EO loaded EOPs are smaller in size than the unloaded PLGA NPs. What is the reason for this decreased size and how to prove the encapsulation of EO in PLGA NPs in this case?   

We cannot provide an explanation for these observations at this point of our studies. It might depend on a different arrangement of the nanoparticles formed by linear PLGA chains in the presence and absence of the loading molecules.

  1. On page 9, lines 307-309, EOs are deposited at a higher volume when loaded in NPs as compared to EO-solo. What is the possible explanation? 

The possible explanation is that the EOs in form of EOLs are entrapped into the lignin matrix. Hence, they are hindered to evaporate by the lignin carrier matrix, an effect and a lignin property cited in literature. In order to underline this explanation we added a phrase including respective references (page 9, lines 312-313).             

  1. The preparation and spray coating method of EO TC/TV-solo is missing in the materials and methods section.

We appreciate the comment and added the respective missing information in the methods section (page 3, lines 138-141).

Round 2

Reviewer 3 Report

Comments and Suggestions for Authors

The authors supplemented the manuscript with more discussions and experimental methods, which makes the description more appropriate. Certain observations cannot be fully explained at this moment, which can be understood.

Regarding comment #1: it is suggested to add the moisture result to the method or discussion section, in order to justify the reason for drying under 40 C for 2 hours as well as proof of sufficient moisture removal at the end of this wood preservation method.

The manuscript can be considered acceptable after this minor revision.

Author Response

Dear reviewer, many thanks. We added a further paragraph in the method section.

Thanks

Manuela Romagnoli